# Cyclic and Randomized Stepsizes
# Invoke Heavier Tails in SGD than Constant Stepsize

**Mert Gürbüzbalaban**                                   *mg1366@rutgers.edu*
*Department of Management Science and Information Systems*
*Rutgers Business School, Piscataway, NJ, USA &*
*Center for Statistics and Machine Learning*
*Princeton University, Princeton, NJ, USA.*

**Yuanhan Hu**                                          *yuanhan.hu@rutgers.edu*
*Department of Management Science and Information Systems*
*Rutgers Business School, Piscataway, NJ, USA*

**Umut Şimşekli**                                       *umut.simsekli@inria.fr*
*Inria, CNRS, Ecole Normale Supérieure*
*PSL Research University, Paris, France*

**Lingjiong Zhu**                                        *zhu@math.fsu.edu*
*Department of Mathematics*
*Florida State University, Tallahassee, FL, USA.*

**Reviewed on OpenReview:** *https://openreview.net/forum?id=lNB5EHx8uC*

## Abstract

Cyclic and randomized stepsizes are widely used in the deep learning practice and can often outperform standard stepsize choices such as constant stepsize in SGD. Despite their empirical success, not much is currently known about when and why they can theoretically improve the generalization performance. We consider a general class of Markovian stepsizes for learning, which contain i.i.d. random stepsize, cyclic stepsize as well as the constant stepsize as special cases, and motivated by the literature which shows that heaviness of the tails (measured by the so-called "tail-index") in the SGD iterates is correlated with generalization, we study tail-index and provide a number of theoretical results that demonstrate how the tail-index varies on the stepsize scheduling. Our results bring a new understanding of the benefits of cyclic and randomized stepsizes compared to constant stepsize in terms of the tail behavior. We illustrate our theory on linear regression experiments and show through deep learning experiments that Markovian stepsizes can achieve even a heavier tail and be a viable alternative to cyclic and i.i.d. randomized stepsize rules.

## 1 Introduction

Stochastic optimization problems are ubiquitous in supervised learning. In particular, the learning problem in neural networks can be expressed as an instance of the stochastic optimization problem

$$\min_{x \in \mathbb{R}^d} \mathbb{E}_{z \sim \mathcal{D}}[f(x, z)], \tag{1}$$

where $x \in \mathbb{R}^d$ are the parameters, $z \in \mathcal{Z}$ is the random data which is assumed to obey an unknown probability distribution $\mathcal{D}$, and $f$ is the loss of misprediction with parameters $x$ corresponding to data $z$. Since the population distribution $\mathcal{D}$ is unknown, in practice, we often approximately solve the empirical version of (1):

$$\min_{x \in \mathbb{R}^d} F(x) := \frac{1}{n} \sum_{i=1}^{n} f_i(x), \quad \text{where} \quad f_i(x) := f(x, z_i), \tag{2}$$

based on a dataset $(z_1, z_2, \ldots, z_n) \in \mathcal{Z}^n$ which consists of independent identically distributed (i.i.d.) samples from data. Stochastic gradient descent (SGD) methods are workhorse methods for solving such problems due to their scalability properties and their favorable performance in practice (Bottou, 2010; 2012). SGD with batch-size $b$ consists of the updates

$$x_{k+1} = x_k - \eta_{k+1} \tilde{\nabla} f(x_k) , \tag{3}$$

where $\eta_{k+1}$ is the stepsize and $\tilde{\nabla} f(x_k) := \frac{1}{b} \sum_{i \in \Omega_k} \nabla f_i(x_k)$ is the stochastic gradient at $k$-th iterate with $\Omega_k$ being a set of data points with cardinality $|\Omega_k| = b$ (see Sec. 2 for details).

Understanding the generalization behavior of SGD, i.e. how a solution found by SGD performs on unseen data has been a major question of research in the last decades (see e.g. Keskar & Socher (2017); Hardt et al. (2016); Lei & Ying (2020); Lin et al. (2016)). Recent empirical and theoretical studies have revealed an interesting phenomenon in this direction. Heavy tails in fact can arise in SGD iterates due to multiplicative noise and the amount of heavy tails (measured by the so-called "tail-index") is significantly correlated with the generalization performance in deep learning practice (Gürbüzbalaban et al., 2021; Hodgkinson & Mahoney, 2021). This phenomenon is not specific to deep learning in the sense that it arises even in surprisingly simple settings such as linear regression (when the loss is a quadratic) with Gaussian input data. This raises the natural question of how the choice of the stepsize sequence $\{\eta_k\}$ affects the tail-index and generalization properties which will be the main topic of study in this work.

The stepsize sequence $\{\eta_k\}$ can be chosen in various ways in a deterministic or a randomized fashion. Constant stepsize as well as varying stepsizes are proposed in the literature (Robbins & Monro, 1951; Kiefer & Wolfowitz, 1952; Smith, 2017; Bottou et al., 2018). For constant stepsize and for (least squares) linear regression problems where the loss $f$ is a quadratic, Gürbüzbalaban et al. (2021) showed that the tails are monotonic with respect to the stepsize and the batch-size and there exists a range of stepsizes for which there exists a stationary distribution with an infinite variance. In a very recent work, Gürbüzbalaban et al. (2022) showed that even heavier tails can arise in the decentralized stochastic gradient descent with constant stepsize and this correlates with generalization. However, other (non-constant) choices of the stepsize can often outperform constant stepsize. Musso (2020) suggests that in the small learning rate regime, uniformly-distributed random learning rate yields better regularization without extra computational cost compared to constant stepsize. Also, cyclic stepsizes where the stepsize changes in a cyclic fashion (obeying lower and upper bounds) have been numerically demonstrated to be very useful for some problems. Among this line of work, Smith (2017) argued that cyclic stepsizes require less tuning in deep learning and can lead to often better performance. Smith (2015) suggests that experiments with sinusoidal learning rates versus cyclic stepsize with equal grid points lead to similar results. Smith & Topin (2017) demonstrates that the cyclical learning rate can produce greater testing accuracy than traditional training despite using large learning rates for different network structures and datasets. Huang et al. (2017) argues that cyclic stepsize allows to train better models compared to constant stepsize. Kalousek (2017) studies a steepest descent method with random stepsizes and shows that it can achieve faster asymptotic rate than gradient descent without knowing the details of the Hessian information. Zhang et al. (2020b) suggests cosine cyclical stepsize for Bayesian deep learning. Gulde et al. (2020) investigates cyclical learning rate and proposes a method for defining a general cyclical learning rate for various deep reinforcement learning problems. They show that using cyclical learning rate achieves similar or even better results than highly tuned fixed learning rates. Wang et al. (2023) applies the cyclical learning rate to train transformer-based neural networks for neural machine translation. They show that the choice of optimizers and the associated cyclical learning rate policy can have a significant impact on performance. Alyafi et al. (2018) argues that with a cyclical learning rate, the neural networks will have better results with similar or even smaller number of epochs on unbalanced datasets without extra computational cost compared to constant stepsize. In addition, due to their popularity, cyclic stepsizes are also part of popular packages such as PyTorch.[1]

Despite the empirical success of randomized stepsize and cyclical stepsize rules, to our knowledge, the effect of the stepsize scheduling (cyclic, randomized versus constant etc.) on the generalization performance is not well studied from a theoretical standpoint except numerical results that highlight the benefits of randomized

---

[1]See e.g. the "CyclicLR package" available at the website https://pytorch.org/docs/stable/generated/torch.optim.lr_scheduler.CyclicLR.html

and cyclic stepsize rules compared to a constant stepsize choice. While studying the generalization itself as a function of the stepsize sequence appears to be a difficult problem; building on the literature which demonstrates that tail-index is correlated with generalization (see e.g. Şimşekli et al. (2020); Barsbey et al. (2021); Şimşekli et al. (2019); Gürbüzbalaban et al. (2021)), we study tail-index as a proxy to generalization performance and demonstrate how the tail-index depends on the stepsize scheduling, highlighting the benefits of using cyclic and randomized stepsizes compared to constant stepsize in terms of the tail behavior.

**Contributions.** We propose a general class of Markovian stepsizes for learning, where the stepsize sequence $\eta_k$ evolves according to a finite-state Markov chain where transitions between the states happen with a certain probability $p \in [0, 1]$. This class recovers i.i.d. random stepsize, cyclic stepsize as well as the constant stepsize as special cases. For losses that have quadratic growth outside a compact set in dimension one, one can infer from the results of Mirek (2011) that SGD iterates with Markovian stepsizes will be heavy-tailed (see Thm. 15 and Thm. 1). However, the assumptions involved are highly nontrivial to check and do not hold for dimension two or higher even for linear regression.[2] Furthermore, these results do neither specify how the tail-index depends on quantities of interest such as the batch-size, dimension nor compare the tails with that of the constant stepsize. Hence, to get finer characterizations of the tail-index and further intuition about the tails, we focus on linear regression as a case study throughout this paper and obtain explicit characterizations that provides further insights as well as theoretical support for the use of cyclic and randomized stepsizes in deep learning practice. We find that even in the linear regression setting, there are significant technical challenges. Here, viewing SGD as an iterated random recursion, we make novel technical contributions in analyzing the iterated random recursions with cyclic and Markovian structure with finite state space. We borrow the idea of regeneration times from probability theory, and consider iterates at those regeneration times to utilize the hidden renewal structure of the underlying iterated random recursions (see Sec. 3 for further details). As a result, we are able to study and characterize the tail-index of the SGD iterates with cyclic stepsizes (Thm. 5), Markovian stepsizes (Thm. 2) as well i.i.d. random stepsizes (Thm. 11).

For cyclic, Markovian and i.i.d. stepsizes, under Gaussian data assumptions for linear regression, we also give a sharp characterization of the tail-index and provide a formula for the range of stepsizes in which SGD iterates admit a stationary distribution with an infinite variance (Lem. 9, Prop. 7, Lem. 11, Prop. 9, Lem. 7, Prop. 6 in the Appendix). We provide non-asymptotic bounds moment bounds for SGD iterates and provide non-asymptotic convergence rates to the stationary distribution in the Wasserstein metric (Thm. 16, Thm. 21, Thm. 14 in the Appendix). These results are obtained by using various technical inequalities, synchronous coupling, and spherical symmetry of Gaussian distributions and are deferred to the Appendix due to space considerations.

In Sec. 4, we compare the tail-indices among these stepsize rules theoretically for linear regression. We show that i.i.d. random stepsizes (where $\eta_k$ are i.i.d. with the same distribution as $\eta$) centered around an expected value $\hat{\eta}$ and a positive range $R$ has heavier tail compared to constant stepsize with $\hat{\eta}$ (Prop. 1) as well as cyclic stepsize centered around $\hat{\eta}$ on a uniform grid with the same range (Prop. 2). Our results show that coarser the grid for the stepsize or wider the stepsize range $R$, heavier the tails become in a sense that we will precise (Thm. 13, Thm. 19, Thm. 20). We prove that reducing the batch-size or increasing the dimension also leads to heavier tails, provided that the tail-index is not too small (Thm. 7, Thm. 4, Thm. 12). These results are obtained by employing delicate analysis leveraging convexity and Jensen's inequality. When the transition probability $p \in (0, 1/2)$ and there are two states, we completely work out the distribution of the regeneration time and the sample path of the Markov chain till the regeneration time to show that Markovian stepsizes can achieve even heavier tail with respect to cyclic and i.i.d. stepsizes and the tail gets heavier if $p$ gets smaller (Prop. 5). We also discuss in Appendix B.4 how our comparisons can be extended to general state spaces. Since smaller tail-indices are correlated with better generalization in deep learning practice, our results shed further light into why random and cyclic stepsizes could perform better than constant stepsize in some deep learning settings. In Sec. 5, we illustrate our theory on linear regression experiments and show through deep learning experiments that our proposed Markovian stepsizes can be a viable alternative to other standard choices in the literature, leading to the best performance in some cases.

---

[2]In Mirek (2011), it is required that the Hessians of the component functions $f(x, z_i)$ have certain orthogonality properties which does not typically hold in linear regression or more generally in statistical learning.

## 2 Technical Background and Literature Review

**Heavy-tailed power-law distributions.** A real-valued random variable $X$ is said to be *heavy-tailed* if the right tail or the left tail of the distribution decays slower than any exponential distribution. A real-valued random variable $X$ is said to have heavy right tail if $\lim_{x \to \infty} \mathbb{P}(X \geq x)e^{cx} = \infty$ for any $c > 0$, and a real-valued random variable $X$ is said to have heavy left tail if $\lim_{x \to \infty} \mathbb{P}(X \leq -x)e^{c|x|} = \infty$ for any $c > 0$. Similarly, an $\mathbb{R}^d$-valued random vector $X$ is said to have heavy tail if $u^T X$ has heavy right tail for some vector $u \in \mathbb{S}^{d-1}$, where $\mathbb{S}^{d-1} := \{u \in \mathbb{R}^d : \|u\| = 1\}$ is the unit sphere in $\mathbb{R}^d$. Heavy tail distributions include $\alpha$-stable distributions, Pareto distribution, log-normal distribution and Weilbull distribution. An important class of the heavy-tailed distributions is the distributions with *power-law* decay, which is the focus of our paper. That is, $\mathbb{P}(X \geq x) \sim c_0 x^{-\alpha}$ as $x \to \infty$ for some $c_0 > 0$ and $\alpha > 0$, where $\alpha$ is known as the *tail-index*, which determines the tail thickness of the distribution. Similarly, the random vector $X$ is said to have power-law decay with tail-index $\alpha$ if for some $u \in \mathbb{S}^{d-1}$, we have $\mathbb{P}(u^T X \geq x) \sim c_0(u)x^{-\alpha}$, for some constant $c_0(u)$ that may depend on the direction $u$ and for some $\alpha > 0$.

**Tail-index in SGD with constant stepsize.** It is possible to show that SGD iterates are heavy-tailed with polynomially-decaying tails admitting a unique stationary distribution, when the loss is strongly convex outside a compact set (Hodgkinson & Mahoney, 2021) or when the loss has linear growth for large $x$ (Mirek, 2011; Gürbüzbalaban et al., 2021) (see Thm. 8 in the Appendix for details), however in these results verifying the assumptions behind are highly non-trivial; and the dependence of the tail-index to the parameters of SGD such as the stepsize is not explicitly given. This motivates the study of least square problems (when $f$ is a quadratic) where more precise characterizations of the tail-index can be obtained (see e.g. (Gürbüzbalaban et al., 2021; Raj et al., 2023a;b; Gürbüzbalaban et al., 2022)). It is also known that slow algorithms that can generalize well subject to heavier tails can be slower for optimization purposes. For example, heavier tails in the gradients and iterates can often result in slower convergence rates in the training process in different optimization settings Şimşekli et al. (2019); Wang et al. (2021), where clipping the gradients and robust versions of stochastic gradient descent have been proposed for achieving faster convergence Zhang et al. (2020a); Gorbunov et al. (2022). Therefore, we believe understanding the heavy-tailedness of the iterates can be beneficial for understanding both optimization and generalization performance of an algorithm, and least square problems serve as a fundamental case where stronger results can be obtained due to their special structure.

Consider the least squares problem $\min_{x \in \mathbb{R}^d} \frac{1}{2n} \|Ax - y\|^2$, where $A$ is an $n \times d$ matrix, with i.i.d. entries, $y \in \mathbb{R}^n$ and $x \in \mathbb{R}^d$. This problem can be written in the form of (2) as

$$F(x) = \frac{1}{n} \sum_{i=1}^n f_i(x), \quad \text{where} \quad f_i(x) := \frac{1}{2} \left(a_i^T x - y_i\right)^2 , \tag{4}$$

where $a_i^T$ is the $i$-th row of the data matrix $A$. Using the fact that $\nabla f_i(x) = a_i \left(a_i^T x - y_i\right)$, we can rewrite SGD iterations (3) as

$$x_{k+1} = \left(I - (\eta_{k+1}/b)H_{k+1}\right)x_k + q_{k+1} , \tag{5}$$

where $H_k := \sum_{i \in \Omega_k} a_i a_i^T$ and $q_k := \frac{\eta_k}{b} \sum_{i \in \Omega_k} y_i$ with $\Omega_k := \{b(k-1)+1, b(k-1)+2, \ldots, bk\}$ and $|\Omega_k| = b$. Here, for simplicity, throughout the paper, we assume that we are in the streaming regime (also called the one-pass setting (Frostig et al., 2015; Jain et al., 2017; Gao et al., 2022)) where each sample is used once and is not recycled. We also make the following assumptions on the data throughout the paper:

**(A1)** $a_i$'s are i.i.d. with a continuous density supported on $\mathbb{R}^d$ with all the moments being finite.

**(A2)** $y_i$ are i.i.d. with a continuous density whose support is $\mathbb{R}$ with all the moments finite.

Assumptions **(A1)** and **(A2)** are satisfied in a large variety of cases, for instance when $a_i$ and $y_i$ are Gaussian distributed. When the stepsize $\eta_k \equiv \eta$ is constant, and $H_k$ are i.i.d., $x_k$ converges to $x_\infty$, where $x_\infty$ has a heavy tail distribution with a tail-index that can be characterized (Gürbüzbalaban et al., 2021). However, to the best of our knowledge, cyclic stepsizes or Markovian stepsizes are not studied in the literature in terms of the tail behavior that they result in the SGD iterates which will be the subject of this work. We also note that decaying stepsize rules where $\eta_k \to 0$ are also widely used and they can ensure convergence of the SGD iterates Sebbouh et al. (2021); Gower et al. (2019). However, in this case, the iterates often converge to a limit point

almost surely and therefore the stationary distribution of the iterates will typically be degenerate as a Dirac mass where characterizing the tail-index of the iterates will no longer be meaningful at stationarity. Therefore, in our analysis, we will assume that the stepsize is bounded away from zero, often lying on a finite grid.

## 3 Main Results

**Stochastic Gradient Descent with Markovian Stepsizes.** We consider Markovian stepsizes with finite state spaces. The discussions with general state spaces will be provided in Section B.4.1 in the Appendix. Let us consider the finite state space

$$\{\eta_1, \eta_2, \ldots, \eta_m, \eta_{m+1}\} = \{c_1, c_2, \ldots, c_{K-1}, c_K, c_{K-1}, \ldots, c_2, c_1\}, \tag{6}$$

where $m = 2K - 2$ and $(c_1, c_2, \ldots, c_K)$ is the stepsize grid. We assume the stepsize goes from $\eta_1$ to $\eta_2$ with probability 1 and it goes from $\eta_K$ to $\eta_{K-1}$ with probability 1. In between, for any $i = 2, 3, \ldots, K-1, K+1, \ldots, m$, the stepsize goes from $\eta_i$ to $\eta_{i+1}$ with probability $p$ and from $\eta_i$ to $\eta_{i-1}$ with probability $1-p$ with the understanding that $\eta_{m+1} := \eta_1$. Therefore, $p = 1$ reduces to the case of cyclic stepsizes. We assume $c_i$ is not the same for every $i$; otherwise this setting reduces to the case of constant stepsizes.

We first observe that SGD (3) is an iterated random recursion of the form

$$x_k = \Psi_k(x_{k-1}, \eta_k), \tag{7}$$

where the map $\Psi_k : \mathbb{R}^d \times \mathbb{R}_+ \to \mathbb{R}^d$, hides the dependence on $\Omega_k$ which are random and i.i.d. and $\eta_k$ are Markovian with the finite state space (6). To the best of our knowledge, there is no general stochastic linear recursion theory for Markovian coefficients, except for some special cases, e.g. with heavy-tailed coefficients (Hay et al., 2011). However, we will show that in the case of finite-state space, it is possible to use the idea of regeneration times from probability theory to analyze the iterated random recursion. The key idea is to introduce the regeneration times $r_k$, which are defined as $r_0 = 0$ and for any $k \geq 1$:

$$r_k := \inf \{j > r_{k-1} : \eta_j = \eta_0\}. \tag{8}$$

That is, $r_k$ are the random times that the stepsizes start at $\eta_0$ at $k = 0$. It is easy to see that $\{r_k - r_{k-1}\}_{k \in \mathbb{N}}$ are i.i.d. with the same distribution as $r_1$. By iterating (7), we have

$$x_{r_{k+1}} = \Psi_{k+1}^{(r)}(x_{r_k}), \tag{9}$$

with $\Psi_{k+1}^{(r)}(\cdot) := \Psi_{r_{k+1}}\left(\cdots\Psi_{r_k+2}(\Psi_{r_k+1}(\cdot, \eta_{r_k+1}), \eta_{r_k+2}), \ldots, \eta_{r_{k+1}}\right)$, where the superscript $(r)$ refers to the random choice of stepsizes and it also happens to be the first letter of "regeneration". Since $\Omega_k$ are i.i.d. over $k$, by the definition of the regeneration time, it follows that $\Psi_k^{(r)}$ are i.i.d. over $k$. We denote that $\Psi_k^{(r)}$ has the common distribution as $\Psi^{(r)}$. If we assume that the random map $\Psi^{(r)}$ is Lipschitz on average, i.e. $\mathbb{E}[L^{(r)}] < \infty$ with $L^{(r)} := \sup_{x,y \in \mathbb{R}^d} \frac{\|\Psi^{(r)}(x) - \Psi^{(r)}(y)\|}{\|x-y\|}$, and is mean-contractive, i.e. if $\mathbb{E}\log(L^{(r)}) < 0$ then it can be shown under further technical assumptions that the distribution of the iterates converges to a unique stationary distribution $x_\infty$ geometrically fast (Diaconis & Freedman, 1999).

The following result can be obtained in dimension $d = 1$, which follows directly from Mirek (2011) by adapting it to our setting (see also Buraczewski et al. (2016)). Mirek (2011) also considers higher dimensions; but for $d > 1$, the assumptions required are not satisfied for quadratic losses nor for least square problems.

**Theorem 1** (Adaptation of Mirek (2011) ). *Assume stationary solution to (12) exists, $d = 1$, and:*

    *(i) There exists a random variable $M^{(r)}$ and a random variable $B^{(r)} > 0$ such that a.s. $|\Psi^{(r)}(x) - M^{(r)}x| \leq B^{(r)}$ for every $x$ where $|\cdot|$ denotes the absolute value;*

    *(ii) The conditional law of $\log|M^{(r)}|$ given $M^{(r)} \neq 0$ is non-arithmetic; i.e. its support is not equal to $a\mathbb{Z}$ for any scalar $a$ where $\mathbb{Z}$ is the set of integers;*

    *(iii) There exists $\alpha^{(r)} > 0$ such that $\mathbb{E}[|M^{(r)}|^{\alpha^{(r)}}] = 1$, $\mathbb{E}[|B^{(r)}|^{\alpha^{(r)}}] < \infty$ and $\mathbb{E}[|M^{(r)}|^{\alpha^{(r)}} \log^+ |M^{(r)}|] < \infty$, where $\log^+(x) := \max(\log(x), 0)$.*

*Then, there exists some constant $c_0^{(r)} > 0$ such that $\lim_{t\to\infty} t^{\alpha^{(r)}} \mathbb{P}(|x_\infty| > t) = c_0^{(r)}$.*

For linear regression, $\Psi^{(r)}$ is a composition of affine maps and stays affine (see our discussion in the proof of Thm. 2); therefore Thm. 1 is applicable in dimension one. More generally, Thm. 1 can be applicable to the restricted class of smooth losses that can be non-convex on a compact while having a quadratic structure outside the compact (so that the gradient is affine up to a constant), and says that heavy tails arises in SGD in this setting but does not precise the tail-index $\alpha^{(r)}$ and this result works only in dimension $d = 1$. This motivates the study of more structured losses in high dimensional settings where more insights can be obtained. When the objective is a quadratic, we will provide a more detailed analysis.

For Markovian stepsizes, we observe that the SGD iterates are given by (5) where $(H_k, q_k)$ is an i.i.d. sequence and $\eta_k$ is a stationary Markov chain with finite state space independent of $(H_k, q_k)_{k \in \mathbb{N}}$. Next, we will show that one can fully characterize the tail-index when the stepsizes follow a Markov chain with a finite state space using a renewal argument based on regeneration times. Let us introduce

$$h^{(r)}(s) := \lim_{k \to \infty} \left( \mathbb{E} \left\| M_k^{(r)} M_{k-1}^{(r)} \ldots M_1^{(r)} \right\|^s \right)^{1/k}, \tag{10}$$

where

$$M_{k+1}^{(r)} := \left( I - (\eta_{r_{k+1}}/b) H_{r_{k+1}} \right) \left( I - (\eta_{r_{k+1}-1}/b) H_{r_{k+1}-1} \right) \cdots \left( I - (\eta_{r_k+1}/b) H_{r_k+1} \right)$$

and $r_k$'s are regeneration times defined in (8). We also define

$$\Pi_k^{(r)} := M_k^{(r)} M_{k-1}^{(r)} \ldots M_1^{(r)},$$

and

$$\rho^{(r)} := \lim_{k \to \infty} (2k)^{-1} \log \left( \text{largest eigenvalue of } \left( \Pi_k^{(r)} \right)^T \left( \Pi_k^{(r)} \right) \right).$$

We have the following result that characterizes the tail-index for the SGD with Markovian stepsizes.

**Theorem 2.** *Consider the SGD iterations (5) with Markovian stepsizes in a finite-state space. If $\rho^{(r)} < 0$ and there exists a unique positive $\alpha^{(r)}$ such that $h^{(r)}\left( \alpha^{(r)} \right) = 1$, then (5) admits a unique stationary solution $x_\infty^{(r)}$ and the SGD iterations with Markovian stepsizes converge to $x_\infty^{(r)}$ in distribution, where the distribution of $x_\infty^{(r)}$ satisfies $\lim_{t \to \infty} t^{\alpha^{(r)}} \mathbb{P}(u^T x_\infty^{(r)} > t) = e_{\alpha^{(r)}}(u)$, for any $u \in \mathbb{S}^{d-1}$, for some positive and continuous function $e_{\alpha^{(r)}}$ on $\mathbb{S}^{d-1}$.*

**Proof.** We recall from (5) that the SGD iterates are given by $x_{k+1} = \left( I - \frac{\eta_{k+1}}{b} H_{k+1} \right) x_k + q_{k+1}$, where $(H_k, q_k)$ is an i.i.d. sequence and $\eta_k$ is a stationary Markov chain with finite state space independent of $(H_k, q_k)_{k \in \mathbb{N}}$. We recall from (8) the regeneration times $r_k$, such that $r_0 = 0$ and for any $k \geq 1$: $r_k := \inf \{ j > r_{k-1} : \eta_j = \eta_0 \}$. That is $r_k$ are the random times that the stepsizes start at $\eta_0$ at $k = 0$. It is easy to see that $\{r_k - r_{k-1}\}_{k \in \mathbb{N}}$ are i.i.d. with the same distribution as $r_1$. It follows that $x_{r_{k+1}} = M_{k+1}^{(r)} x_{r_k} + q_{k+1}^{(r)}$, where $M_{k+1}^{(r)}$ and $q_{k+1}^{(r)}$ are defined as:

$$M_{k+1}^{(r)} := \left( I - (\eta_{r_{k+1}}/b) H_{r_{k+1}} \right) \left( I - (\eta_{r_{k+1}-1}/b) H_{r_{k+1}-1} \right) \cdots \left( I - (\eta_{r_k+1}/b) H_{r_k+1} \right),$$

$$q_{k+1}^{(r)} := \sum_{i=r_k}^{r_{k+1}} \left( I - (\eta_{r_{k+1}}/b) H_{r_{k+1}} \right) \left( I - (\eta_{r_{k+1}-1}/b) H_{r_{k+1}-1} \right) \cdots \left( I - (\eta_{i+1}/b) H_{i+1} \right) q_i.$$

Since $r_k$ are regeneration times, one can easily check that $\left( M_k^{(r)}, q_k^{(r)} \right)$ are i.i.d. in $k$. The rest of the proof follows from Theorem 4.4.15 in Buraczewski et al. (2016) which goes back to Theorem 1.1 in Alsmeyer & Mentemeier (2012) and Theorem 6 in Kesten (1973). See also Goldie (1991); Buraczewski et al. (2015). The proof is complete. $\qquad \square$

**Remark 1.** *It is possible to extend some of our results given in Thm. 2 for linear regression to problems where the loss function is a convex quadratic up to an error term outside a compact set. Smoothed Lasso problems where the penalty term is a smooth version of the $\ell_1$ penalty would be an example. In this more general case, while it is not possible to provide a formula for the tail-index exactly, one can still provide lower and upper bounds on the tail-index. The details can be found in Appendix F.*

In general, there is no simple explicit formula for $h^{(r)}(s)$ to evaluate $\alpha^{(r)}$. However, we can easily obtain the following bound using the sub-multiplicativity of the norm of matrix products:

$$h^{(r)}(s) \leq \hat{h}^{(r)}(s) := \mathbb{E}\left[\prod_{i=1}^{r_1} \|I - (\eta_i/b)H_i\|^s\right] = \mathbb{E}\left[\prod_{i=1}^{r_1} \mathbb{E}_H\left[\|I - (\eta_i/b)H\|^s\right]\right], \tag{11}$$

where $\mathbb{E}_H$ denotes the expectation taken over $H_i$, which are i.i.d. distributed as $H$, and which are independent of $(\eta_k)_{k\in\mathbb{N}}$, where $r_1$ is the regeneration time defined in (8). We define the lower bound $\hat{\alpha}^{(r)}$ for the tail-index $\alpha^{(r)}$ as the unique positive value such that $\hat{h}^{(r)}\left(\hat{\alpha}^{(r)}\right) = 1$, provided that $\hat{\rho}^{(r)} := \mathbb{E}\left[\sum_{i=1}^{r_1} \mathbb{E}_H\left[\log\|I - \frac{\eta_i}{b}H\|\right]\right] < 0$. In the following, we show that the lower bound $\hat{\alpha}^{(r)}$ for the tail-index is increasing in batch-size.

**Theorem 3.** *$\hat{\alpha}^{(r)}$ is strictly increasing in batch-size $b$ provided that $\hat{\alpha}^{(r)} \geq 1$.*

Our proof of Theorem 3 is based on the fact that the function $h^{(r)}(s)$ is strictly decreasing in $b$ for $s \geq 1$ which follows from Jensen's inequality and convexity of the function $\|\cdot\|^s$ for $s \geq 1$. The function $\|\cdot\|^s$ is not convex for $s < 1$; this is the reason why the condition $\hat{\alpha}^{(r)} \geq 1$ is needed in Theorem 3 within our analysis. Note also that the tail-index $\alpha^{(r)}$ is the unique positive value such that $h^{(r)}\left(\alpha^{(r)}\right) = 1$ provided that $\rho^{(r)} < 0$. In general, there is no simple closed-form expression for $h^{(r)}(s)$ that is defined in (10).

However, when the input data $a_i$ are Gaussian, we are able to obtain more explicit expression for $h^{(r)}(s)$.

**(A3)** $a_i \sim \mathcal{N}(0, \sigma^2 I_d)$ are Gaussian distributed for every $i$.

Under **(A3)**, we can obtain a more explicit expression (see Lem. 11 and Lem. 12 in the Appendix) to characterize the tail-index $\alpha^{(r)}$. Under **(A3)**, we obtain the following result that shows the tail-index $\alpha^{(r)}$ is increasing in batch-size $b$ and decreasing in dimension $d$.

**Theorem 4.** *Assume (A3) holds and $\rho^{(r)} < 0$. Then: (i) The tail-index $\alpha^{(r)}$ is strictly increasing in batch-size $b$ provided that $\alpha^{(r)} \geq 1$; (ii) The tail-index $\alpha^{(r)}$ is strictly decreasing in dimension $d$.*

**Stochastic Gradient Descent with Cyclic Stepsizes.** As a special case of the Markovian stepsizes, we consider the stochastic gradient descent method with cyclic stepsizes. More specifically, we assume that $\eta_k$ takes values on a grid $(c_1, c_2, \ldots, c_K)$ in a cyclic manner satisfying (6) and the length of the cycle is $m = 2K - 2$. In other words $\eta_{mk+i} = \eta_i$, $i = 1, 2, \ldots, m$, for any $k = 0, 1, 2, \ldots$ Note that we can view cyclic stepsizes as a special case of Markovian stepsizes with transition probability $p = 1$.

As a special case of the Markovian stepsizes, the SGD (3) can be iterated and we can consider:

$$x_{(k+1)m} = \Psi_{k+1}^{(m)}(x_{km}), \tag{12}$$

where $\Psi_{k+1}^{(m)}(\cdot) := \Psi_{(k+1)m}\left(\cdots \Psi_{km+2}(\Psi_{km+1}(\cdot, \eta_{km+1}), \eta_{km+2}), \ldots, \eta_{(k+1)m}\right)$ are i.i.d. over $k$. This is basically the map that corresponds to consecutive $m$ iterations of SGD, which demonstrates an i.i.d. structure. We denote $\Psi^{(m)}$ as the common distribution of $\Psi_k^{(m)}$ where the superscript $(m)$ highlights the dependence on the cycle length $m$. In this case, an analogue of Thm. 1 can be obtained in dimension one (see Thm. 15 in the Appendix); but as before in the rest of the discussion, we focus on the quadratic case to have finer results for the tail-index. We recall the SGD iterates from (5), where we consider the stepsize $\eta_k$ to be deterministic and cyclic with a cycle length $m$. Next, let us introduce

$$h^{(m)}(s) := \lim_{k\to\infty}\left(\mathbb{E}\left\|M_k^{(m)}M_{k-1}^{(m)}\ldots M_1^{(m)}\right\|^s\right)^{1/k}, \tag{13}$$

where

$$M_k^{(m)} := \left(I - (\eta_m/b)H_{km}\right)\left(I - (\eta_{m-1}/b)H_{km-1}\right)\cdots\left(I - (\eta_1/b)H_{(k-1)m+1}\right), \tag{14}$$

is the product of consecutive $m$ iteration matrices. We also define

$$\rho^{(m)} := \lim_{k\to\infty}(2k)^{-1}\log\left(\text{largest eigenvalue of } \left(\Pi_k^{(m)}\right)^T\left(\Pi_k^{(m)}\right)\right), \quad \Pi_k^{(m)} := M_k^{(m)}M_{k-1}^{(m)}\ldots M_1^{(m)}.$$

We can iterate the SGD from (5) to obtain $x_{(k+1)m} = M_{k+1}^{(m)} x_{km} + q_{k+1}^{(m)}$, where $M_{k+1}^{(m)}$ is defined in (14) and $q_{k+1}^{(m)} := \sum_{i=km+1}^{(k+1)m} \left(I - \frac{\eta_{(k+1)m}}{b} H_{(k+1)m}\right) \left(I - \frac{\eta_{(k+1)m-1}}{b} H_{(k+1)m-1}\right) \cdots \left(I - \frac{\eta_{i+1}}{b} H_{i+1}\right) q_i$. We have the following result that characterizes the tail-index.

**Theorem 5.** *Consider the SGD iterations (5) with cyclic stepsizes $\{\eta_k\}$ where the length of the cycle is $m$. If $\rho^{(m)} < 0$ and there exists a unique positive $\alpha^{(m)}$ such that $h^{(m)}\left(\alpha^{(m)}\right) = 1$, then (5) admits a unique stationary solution $x_\infty^{(m)}$ and the SGD iterations with cyclic stepsizes converge to $x_\infty^{(m)}$ in distribution, where the distribution of $x_\infty^{(m)}$ satisfies $\lim_{t \to \infty} t^{\alpha^{(m)}} \mathbb{P}(u^T x_\infty^{(m)} > t) = e_{\alpha^{(m)}}(u)$, for any $u \in \mathbb{S}^{d-1}$, for some positive and continuous function $e_{\alpha^{(m)}}$ on $\mathbb{S}^{d-1}$.*

**Proof.** This follows immediately from Theorem 2 by noting that Markovian stepsizes reduce to the cyclic stepsizes in the special case when $p = 1$. $\qquad\square$

Determining the exact value of the tail-index $\alpha^{(m)}$ for the stationary distribution $x_\infty^{(m)}$ seems to be a hard problem; nevertheless, we can characterize a lower bound for the tail-index to control how heavy tailed SGD iterates can be by following a similar approach to our discussions for the Markovian stepsizes. We start by noticing that $h^{(m)}(s) = 1$ if and only if $\left(h^{(m)}(s)\right)^{1/m} = 1$. In general, there is no simple explicit formula for $h^{(m)}(s)$. However, we have the following bound due to the sub-multiplicativity of the norm of matrix products:

$$\left(h^{(m)}(s)\right)^{1/m} \leq \hat{h}^{(m)}(s) := \left(\prod_{i=1}^m \mathbb{E}\left[\left\|I - \frac{\eta_i}{b} H\right\|^s\right]\right)^{1/m}.$$

Let $\hat{\alpha}^{(m)}$ be the unique positive value such that $\hat{h}^{(m)}\left(\hat{\alpha}^{(m)}\right) = 1$ provided that $\hat{\rho}^{(m)} := \sum_{i=1}^m \mathbb{E}\left[\log\left\|I - \frac{\eta_i}{b} H\right\|\right] < 0$. This provides a lower bound for the tail-index $\alpha^{(m)}$. Next, we show that the lower bound $\hat{\alpha}^{(m)}$ for tail-index is increasing in batch-size. We next state analogues of Thm. 3 and Thm. 4 in the cyclic stepsize setting, the proofs are similar, with the only difference that regeneration times (the time it takes to revisit a particular stepsize) are random for Markovian stepsizes, whereas they are deterministic for cyclic stepsizes. The proofs are given in the appendix for the sake of completeness.

**Theorem 6.** *$\hat{\alpha}^{(m)}$ is strictly increasing in batch-size $b$ provided that $\hat{\alpha}^{(m)} \geq 1$.*

Under **(A3)**, we can get a more explicit formula for $h^{(m)}(s)$ and $\rho^{(m)}$ (see Lem. 9, Lem. 10 in the Appendix) and hence can obtain further properties of the tail-index $\alpha^{(m)}$. Under **(A3)**, we obtain the monotonic dependence of the tail-index $\alpha^{(m)}$ on the batch-size $b$ and the dimension $d$.

**Theorem 7.** *Assume **(A3)** holds and $\rho^{(m)} < 0$. Then: (i) The tail-index $\alpha^{(m)}$ is strictly increasing in batch-size $b$ provided $\alpha^{(m)} \geq 1$. (ii) The tail-index $\alpha^{(m)}$ is strictly decreasing in dimension $d$.*

## 4  Comparisons of Tail-Indices

In this section, we compare the tail-indices of SGD with constant, i.i.d., cyclic and Markovian stepsizes. The i.i.d. stepsizes can be considered as a special case of the Markovian stepsizes and in Sec. B.2 in the Appendix, we study the SGD with i.i.d. stepsizes in detail. Under Assumption **(A3)**, we compare the tail-index $\alpha$ of the SGD with i.i.d. stepsizes (where $\eta_k = \eta$ has the same distribution for every $k$) with the SGD with constant stepsize (where stepsize is fixed at $\mathbb{E}[\eta]$).

**Proposition 1.** *Assume **(A3)** holds and $\rho < 0$. Then the tail-index $\alpha$ with i.i.d. stepsize is strictly less than the tail-index $\alpha_c$ with constant stepsize $\mathbb{E}[\eta]$ provided $\alpha \geq 1$.*

In Prop. 1 we showed that the tail-index $\alpha$ is strictly less than the tail-index $\alpha_c$ with constant stepsize $\mathbb{E}[\eta]$. Under **(A3)**, we can also compare the tail-index $\alpha^{(m)}$ of the SGD with cyclic stepsizes and the tail-index $\alpha$ of the SGD with i.i.d. uniformly distributed stepsizes such that $\mathbb{P}(\eta = \eta_i) = \frac{1}{m}$, for any $1 \leq i \leq m$. Then, we have the following result which says i.i.d. stepsizes have heavier tail.

**Proposition 2.** *Assume **(A3)** holds. The tail-index $\alpha$ for the SGD with i.i.d. stepsizes such that $\mathbb{P}(\eta = \eta_i) = \frac{1}{m}$ for any $1 \leq i \leq m$ is smaller than the tail-index $\alpha^{(m)}$ for SGD with cyclic stepsizes.*

Next, we compare the the tail-index $\alpha^{(m)}$ with cyclic stepsizes and the tail-index $\alpha_c$ with constant stepsize $\frac{1}{m}\sum_{i=1}^{m}\eta_i$. When the batch-size is not too large relative to the dimension (i.e. when $d \geq b + 3$), we can show that cyclic stepsizes lead to heavier tails under **(A3)**. The proof is based on exploiting log-convexity properties of the $h^{(m)}(\cdot)$ function when the batch-size is in this regime.

**Proposition 3.** *Assume **(A3)** holds and $d \geq b+3$. Then the tail-index $\alpha^{(m)}$ with cyclic stepsizes is strictly smaller than the tail-index $\alpha_c$ with constant stepsize $\frac{1}{m}\sum_{i=1}^{m}\eta_i$ provided that $\eta_i$'s are sufficiently small.*

In general, for SGD with Markovian stepsizes, the regeneration time is hard to analyze. Next, we consider the simplest example of a Markov chain, i.e., a homogeneous Markov chain with two-state space $\{\eta_l, \eta_u\}$ such that $\mathbb{P}(\eta_1 = \eta_u | \eta_0 = \eta_l) = p$ and $\mathbb{P}(\eta_1 = \eta_l | \eta_0 = \eta_u) = p$. This Markov chain exhibits a unique stationary distribution $\mathbb{P}(\eta_0 = \eta_\ell) = \mathbb{P}(\eta_0 = \eta_u) = \frac{1}{2}$. We notice that the special case $p = 1$ reduces to the cyclic stepsizes. Indeed, we have the following monotonicity result of the tail-index depending on the parameter $p$.

**Proposition 4.** *Consider the two-state Markov chain, i.e. $\mathbb{P}(\eta_1 = \eta_u | \eta_0 = \eta_l) = p$ and $\mathbb{P}(\eta_1 = \eta_l | \eta_0 = \eta_u) = p$ and assume that $p \in \mathcal{P}$, where*

$$\mathcal{P} := \left\{ p \in [0,1] : (1-p)\max\left(\mathbb{E}_H\left[\left\|\left(I - \frac{\eta_l}{b}H\right)e_1\right\|^{\alpha^{(r)}}\right], \mathbb{E}_H\left[\left\|\left(I - \frac{\eta_u}{b}H\right)e_1\right\|^{\alpha^{(r)}}\right]\right) < 1 \right\}. \quad (15)$$

*Then, the tail-index $\alpha^{(r)}$ is increasing in $p \in \mathcal{P}$. In particular, $\alpha^{(r)} \leq \alpha^{(m)}$.*

Prop. 4 shows that Markovian stepsizes lead to heavier tails than cyclic stepsizes. In Prop. 2, we showed that SGD with i.i.d. stepsizes lead to heavier tail than the SGD with cyclic stepsizes, which has heavier tail than the SGD with constant stepsizes (Prop. 3). The next result states that Markovian stepsizes may lead to heavier tails than i.i.d. stepsizes depending on whether $p$ is greater or less than $\frac{1}{2}$.

**Proposition 5.** *Under the setting of Propositions 3 and 4, we have $\alpha < \alpha^{(r)} < \alpha^{(m)} < \alpha_c$ for any $\frac{1}{2} < p < 1$ and we have $\alpha^{(r)} < \alpha < \alpha^{(m)} < \alpha_c$ for any $p < \frac{1}{2}$, where $\alpha_c$, $\alpha$, $\alpha^{(m)}$, $\alpha^{(r)}$ denote the tail-index for SGD with constant, i.i.d., cyclic and Markovian stepsizes respectively.*

## 5 Numerical Experiments

In this section, we present the numerical experiments including the tail-index estimation of linear regression using uniform stepsize and Markovian stepsize and the performance of uniform, Markovian, cyclic and constant stepsize on deep learning settings.

**Linear regression (least squares).** In the following experiments, we investigate the relationship between the tail-index and the stepsize choice. We consider the following model: $w \sim \mathcal{N}\left(0, \sigma^2 I\right)$, $x_i \sim \mathcal{N}\left(0, \sigma_x^2 I\right)$, and $y_i | w, x_i \sim \mathcal{N}\left(x_i^T w, \sigma_y^2\right)$, where $w, x_i \in \mathbb{R}^d$, $y_i \in \mathbb{R}$ for $i = 1, \ldots, n$, and $\sigma, \sigma_x, \sigma_y > 0$. In the experiments, we set $d = 100$, take $\sigma = 3, \sigma_x = 1, \sigma_y = 3$, and generate $\{x_i, y_i\}_{i=1}^n$ by simulating the statistical model. To estimate the stationary measure, we iterate SGD for 1000 iterations and we repeat this for 10000 runs. At each run, we take the average of the last 500 SGD iterates; in the Appendix (Cor. 1, 2 and 3) we argue that the average follows an $\alpha$-stable distribution under some assumptions. This enables us to use advanced estimators (Mohammadi et al., 2015) specific to $\alpha$-stable distributions. For constant stepsize schedules, we set the stepsize to be constant $\hat{\eta}$ during the training process. For i.i.d. uniformly distributed stepsize, cyclic stepsize, and Markovian stepsize, the interval of stepsize is set as $[\hat{\eta} - R, \hat{\eta} + R]$.

To compare different stepsize (learning rate) schedules, we calculate the tail-index for the linear regression experiment in Fig. 1(a). In this experiment, we set batch-size 10 for all stepsize schedules, $R = 0.05$ and $K = 10$ for uniform, cyclic, and Markovian stepsize schedules. For Markovian stepsize, we set $p = 0.6$. We can conclude from the figure that the tail-index of all these stepsize schedules will decrease when the mean of stepsize $\hat{\eta}$ increases, meanwhile the tail-index of SGD with i.i.d. uniformly distributed stepsize is smaller than the tail-index of cyclic stepsize schedule, which validate our Prop. 2. We can also observe that the tail-index of Markovian stepsize is smaller than the cyclic stepsize schedule, which is predicted by our Prop. 5.

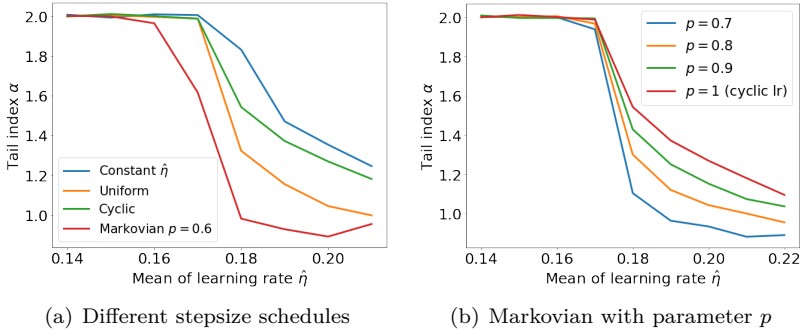

(a) Different stepsize schedules    (b) Markovian with parameter $p$

Figure 1: Center of the stepsize interval ($\hat{\eta}$) vs. tail-index for linear regression. (Left panel) Comparison of constant, uniform, cyclic and Markovian stepsizes with $p = 0.6$ with batch-size $b = 10$, and the stepsize grid parameters $R = 0.05$ and $K = 10$. (Right panel) Comparison of Markovian stepsizes with different transition probabilities $p$ when the mean of the learning rate $\hat{\eta}$ is varied, with batch-size $b = 10$, $R = 0.05$.

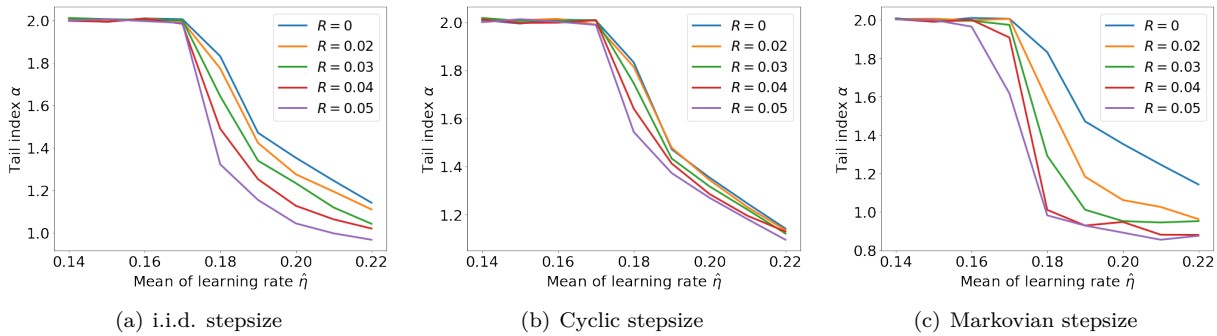

(a) i.i.d. stepsize    (b) Cyclic stepsize    (c) Markovian stepsize

Figure 2: Tail-index corresponding to different range values ($R$) as a function of the mean stepsize $\hat{\eta}$ for linear regression, for i.i.d. random stepsizes (left panel), for cyclic stepsize (middle panel) and for Markovian stepsizes with $p = 0.6$ (right panel). We fix $K = 10$ and vary $\delta$ from 0 to 0.05.

In the next sets of experiments, we will investigate the influence of different values of parameters on the tail-index for different stepsize schedules. In the first experiment, we test different transition probability $p$ for Markovian stepsize. The result is shown in Fig. 1(b), where the values of $p$ is varied from 0.6 to 1 where we set batch-size $b = 10$, $R = 0.05$. When $p = 1$, the Markovian stepsize schedule will degenerate to cyclic stepsize. We observe that with a smaller $p$ value, the tail-index of Markovian stepsize will become smaller. This result validates our Prop. 4. In another experiment, we test different range $R$ values for i.i.d. uniformly distributed, cyclic and Markovian stepsize schedules where we set the batch-size $b = 10$ and $p = 0.6$ for Markovian stepsize. To vary $R = (K - 1)\delta/2$, we keep $K = 10$ as a constant and vary $\delta$ from 0 to 0.05. As shown in Fig. 2, the tail-index of all schedules will decrease when the range $R$ increases. These results validate our Thm. 19 and Thm. 13 in the Appendix. Finally, in Fig. 3, we test different batch-size values from 5 to 15 where we set range $R = 0.05$ and for Markovian stepsize $p = 0.6$. We observe that the tail-index of every stepsize schedule is strictly increasing in batch-size, which is consistent with our theory (Thm. 7, 4 and Thm. 12 in the Appendix).

**Deep Learning.** In our second set of experiments, we investigate the performance of uniform stepsize, Markovian stepsize, cyclic stepsize, and constant stepsize beyond the linear regression setting. Here we consider the 3-layer fully connected network with the cross entropy loss on the MNIST dataset. We train the models using SGD with batch-size $b = 20$. Similar to the linear regression setting, we test different stepsizes centered at $\hat{\eta}$ where we vary $\hat{\eta}$ while keeping the range of stepsize fixed with $R = 0.05$ and $K = 100$ for cyclic and Markovian stepsize schedules. We display the results of the deep learning experiment in Table 1. Following the literature (Chen et al., 2018), we measure generalization in terms of the difference between the

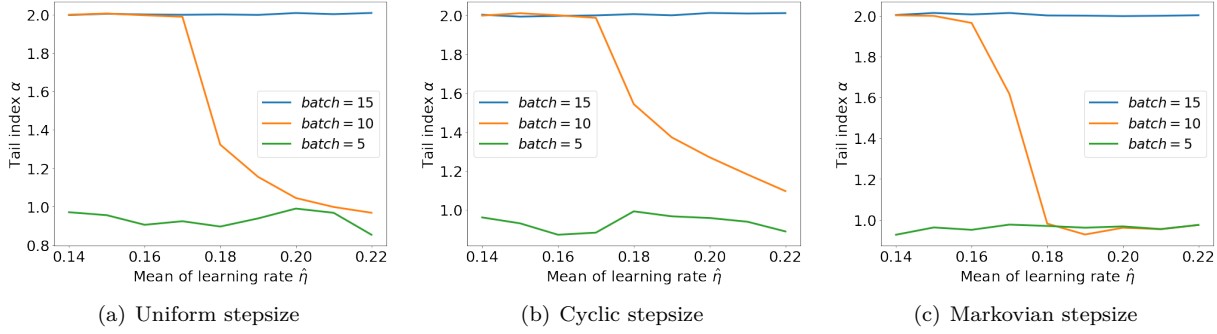

Figure 3: Tail-index of batch-size value for linear regression for uniform stepsize (left panel), cyclic stepsize (middle panel) and Markovian stepsize with $p = 0.6$ (right panel). We test different batch-size values $b \in \{5, 10, 15\}$ with range parameter $R = 0.05$ and $K = 10$.

training and test loss; the smaller difference is the better generalization. As an alternative but correlated metric, we also consider the difference between test and training accuracy for quantifying the generalization performance. For different center stepsize $\hat{\eta}$, we can observe that uniform, cyclic, and Markovian stepsize schedules lead to a smaller tail-index compared to the constant stepsize. While the tail-index and heavy tail are well correlated according to existing theoretical and numerical results (Şimşekli et al., 2020; Barsbey et al., 2021; Raj et al., 2023a;b; Gürbüzbalaban et al., 2021), the relationship is not a perfect straight line. For example, we can see that for $\hat{\eta} = 0.06$ case, the smallest tail-index does not lead to the better generalization power in the sense of smaller error difference and accuracy difference. But for all the other cases where $\hat{\eta} = 0.08, 0.09$, Markovian stepsize schedule with $p = 0.5$ has both the smallest tail-index and best generalization performance.

As a larger-scale deep learning experiment, we next consider the VGG-11 architecture that has 8 convolutional layers and 3 fully-connected layers and we use the CIFAR10 dataset. For the stepsize grid parameters, we use $R = 0.03$ and $K = 100$ and take batchsize $b = 25$. We use SGD for training and compare different stepsize schedules in Table 2 for different choices of the mean stepsize $\hat{\eta}$, similarly to the previous experiment. We observe that i.i.d. stepsize has the heaviest tails (with the lowest tail index) in all cases, and the constant stepsize has the lightest tails (with the highest tail index) whereas the cyclic and Markovian stepsizes are in between except when $\hat{\eta} = 0.12$. As a general trend, these results are roughly inline with our results given in Prop. 5 that compares the tails of different stepsizes schedules under some assumptions. In Figure 2, we also see that the heaviest tails led to the best generalization in terms of accuracy, except for the $\hat{\eta} = 0.11$ case. It would be interesting to investigate how the tails are related to generalization and optimization performance further as a part of future work.

To summarize, our results suggest that good performance associated to cyclic, randomized and Markovian stepsize can be due to the incurrence of heavier tails compared to constant stepsize in the deep learning settings. While deep learning setting is significantly more complicated than the linear regression setting we considered in our theoretical results, our results offer theoretical support into why alternative stepsizes (randomized, cyclic) can be successful and offers new Markovian stepsize rules that can perform better in some cases.

## 6 Conclusion

In this work, we proposed Markovian stepsizes which recovers uniformly random, cyclic and constant stepsizes as special cases. We developed proof techniques where we show that uniformly random, cyclic and Markovian stepsizes can lead to heavier tails in the distribution of SGD iterates. Since smaller tail-indices are correlated with better generalization in deep learning practice, our results shed further light into why random and cyclic stepsizes can perform better than constant stepsize in deep learning. We also showed that our proposed

| Schedule | $\hat{\eta}$ | Train Err | Test Err | Error diff | Train Acc | Test Acc | Acc diff | Tail-index |
|---|---|---|---|---|---|---|---|---|
| Constant | | 3.81E-07 | 0.0055 | **5.54E-03** | 100% | 98.39% | **1.61%** | 1.95 |
| Uniform | | 1.70E-07 | 0.0067 | 6.69E-03 | 100% | 98.38% | 1.62% | 1.93 |
| Cyclic | 0.06 | 3.27E-07 | 0.0065 | 6.50E-03 | 100% | 98.33% | 1.67% | 1.98 |
| Markovian $p$=0.6 | | 2.97E-07 | 0.0062 | 6.16E-03 | 100% | 98.36% | 1.64% | 1.90 |
| Markovian $p$=0.5 | | 2.79E-07 | 0.0068 | 6.84E-03 | 100% | 98.34% | 1.66% | **1.90** |
| Constant | | 0.00194 | 0.0332 | 3.13E-02 | 99.55% | 97.64% | 1.91% | 1.95 |
| Uniform | | 0.00221 | 0.0346 | 3.24E-02 | 99.59% | 97.63% | 1.96% | 1.94 |
| Cyclic | 0.08 | 0.01490 | 0.0346 | 1.97E-02 | 94.99% | 93.17% | 1.82% | 1.78 |
| Markovian $p$=0.6 | | 0.00055 | 0.0198 | 1.93E-02 | 99.76% | 97.80% | 1.96% | 1.89 |
| Markovian $p$=0.5 | | 2.10E-07 | 0.0071 | **7.14E-03** | 100% | 98.35% | **1.65%** | **1.69** |
| Constant | | 0.00154 | 0.0366 | 3.50E-02 | 99.61% | 97.71% | 1.90% | 1.85 |
| Uniform | | 0.00208 | 0.0283 | 2.62E-02 | 99.49% | 97.48% | 2.01% | 1.77 |
| Cyclic | 0.09 | 0.00294 | 0.0316 | 2.87E-02 | 99.18% | 97.22% | 1.96% | 1.62 |
| Markovian $p$=0.6 | | 0.00433 | 0.0266 | 2.22E-02 | 99.04% | 97.05% | 1.99% | 1.69 |
| Markovian $p$=0.5 | | 0.00168 | 0.0217 | **2.00E-02** | 99.38% | 97.52% | **1.86%** | **1.51** |

Table 1: 3-layer fully connected network on the MNIST dataset. We vary the mean of the stepsize $\hat{\eta}$ and compare the stepsize schedules, where in the third to nineth columns we report the training error, test error, difference between the test and training errors, training accuracy, test accuracy, difference between the test and training accuracy and the tail-index respectively.

| Schedule | $\hat{\eta}$ | Train Err | Test Err | Error diff | Train Acc | Test Acc | Acc diff | Tail-index |
|---|---|---|---|---|---|---|---|---|
| Constant | | 3.49E-08 | 0.012 | 1.18E-02 | 1 | 85.15% | 14.85% | 1.83 |
| Uniform | 0.10 | 3.38E-08 | 0.011 | 1.15E-02 | 1 | 85.19% | **14.81%** | **1.80** |
| Cyclic | | 3.81E-08 | 0.011 | **1.14E-02** | 1 | 84.69% | 15.31% | 1.82 |
| Markovian $p$=0.7 | | 2.81E-08 | 0.012 | 1.18E-02 | 1 | 85.05% | 14.95% | 1.83 |
| Constant | | 5.59E-08 | 0.011 | **1.14E-02** | 1 | 85.21% | **14.79%** | 1.84 |
| Uniform | 0.11 | 3.39E-08 | 0.012 | 1.20E-02 | 1 | 84.77% | 15.23% | **1.79** |
| Cyclic | | 2.74E-08 | 0.012 | 1.17E-02 | 1 | 84.91% | 15.09% | 1.82 |
| Markovian $p$=0.7 | | 3.24E-08 | 0.012 | 1.16E-02 | 1 | 85.13% | 14.87% | 1.79 |
| Constant | | 3.99E-08 | 0.012 | 1.18E-02 | 1 | 85.33% | 14.67% | 1.80 |
| Uniform | 0.12 | 6.85E-08 | 0.012 | 1.21E-02 | 1 | 85.41% | **14.59%** | **1.76** |
| Cyclic | | 4.25E-08 | 0.012 | **1.16E-02** | 1 | 85.15% | 14.85% | 1.80 |
| Markovian $p$=0.7 | | 3.02E-08 | 0.013 | 1.29E-02 | 1 | 84.57% | 15.43% | 1.81 |

Table 2: VGG11 network on the CIFAR10 dataset. We vary the mean of the stepsize $\hat{\eta}$ and compare the stepsize schedules, where in the third to nineth columns we report the training error, test error, difference between the test and training errors, training accuracy, test accuracy, difference between the test and training accuracy and the tail-index respectively.

Markovian stepsizes can be a viable alternative to other standard choices in the literature, leading to the best performance in some cases.

**Acknowledgments**

Mert Gürbüzbalaban and Yuanhan Hu acknowledge Rutgers Business School for creating a supportive research atmosphere, most of this work was completed when Yuanhan Hu was a Ph.D. student at the Rutgers Business School. Mert Gürbüzbalaban and Yuanhan Hu's research are supported in part by the grants Office of Naval Research Award Number N00014-21-1-2244, National Science Foundation (NSF) CCF-1814888, NSF DMS-2053485. Umut Şimşekli's research is supported by the French government under management of Agence Nationale de la Recherche as part of the "Investissements d'avenir" program, reference ANR-19-P3IA-0001 (PRAIRIE 3IA Institute) and the European Research Council Starting Grant DYNASTY – 101039676. Lingjiong Zhu is grateful to the partial support from a Simons Foundation Collaboration Grant and the grant NSF DMS-2053454 from the National Science Foundation.

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
