# OpenReview forum: "Cyclic and Randomized Stepsizes Invoke Heavier Tails in SGD than Constant Stepsize"
_TMLR — Accepted by TMLR_

### Review · Reviewer_BHTt · 2023-07-12

**Summary Of Contributions:**

This paper studies the tail effect of using cyclic, random, and constant step-sizes in the learning process of running SGD on the linear regression problem. The authors have shown that using these special (cyclic, random, etc) step-sizes can actually lead to heavier tails in the distribution of SGD iterates, compared with using constant step-sizes, which is related to better generalization performance. Most of the work is theoretical, but the authors also provide experimental evidence to support their claims.

**Audience:**

Yes

**Broader Impact Concerns:**

N.A.

**Claims And Evidence:**

Yes

**Requested Changes:**

1. The current theoretical analysis focuses on purely linear regression, which is not bad, but I still wonder what the true barrier is if we want to migrate the analysis to neural networks. After many years of development in Neural Tangent Kernel theory, I think the analysis of neural networks should be similar to that of linear regression, especially if we add the assumption that the network can be infinitely wide. See, e.g., https://arxiv.org/abs/1909.12292. Could the authors at least provide some insights on what the true barriers are if we try to conduct a similar analysis in deep learning?

2. The whole section 3 is a little hard to read and takes me a lot of time to understand the notations and proofs. It seems to be an unnecessary burden since the theorems and Section 4 convey the most important information. I would really suggest polishing this section to make it easier for the readers.

3. In Section 4, I wonder what the relationship is between the tail-index of cyclic step-sizes and constant step-sizes. It seems that the comparison is missing. Please correct me if I'm wrong.

4. In the experiments, the authors have only chosen linear regression and 3-layer MLP as the baseline model. I would suggest adding one more deep neural network experiments, e.g., ResNet50, Transformers.

5. The captions of all the figures are highly un-informative and I would suggest adding more related info of the figures to these captions. For example, the experiment setting (linear regression or deep learning), parameter settings, what each line represents and so on.

**Strengths And Weaknesses:**

Strengths:
1. The study of using cyclic, random, and even Markovian step sizes and their effect in learning is, as far as I know, novel and interesting. The industrial side has been using different step size designs for a long time. This study could be a good theoretical understanding of why these tricks are useful.

2. The authors have provided a thorough and detailed analysis of the tail-indices. Although I don't have the time to check the proofs, the theoretical results look promising and are supported by the experimental results.

3. The literature review and the introduction is more than complete and interesting.

Weaknesses:
1. The theoretical study focuses on purely linear regression and not deep learning models.

2. The paper is a little hard to read because of the heavy notation burden.

3. Some of the assumptions seem to be too strong to be true, especially for deep learning models.

---

### Review · Reviewer_Ft9k · 2023-07-13

**Summary Of Contributions:**

Put roughly, the focus of this paper is the following question: "how do properties of stochastic gradient-based learning algorithms impact the distribution of the resulting iterates?" The context for this question is a rather substantial literature over the past few years which draws connections between better generalization and iterates with "lighter tails".

Of the "properties" of learning algorithms just mentioned, the particular focus of this paper is on step size strategies, allowed to be stochastic, albeit independent of the data. The main contributions here are theoretical in nature. They consider stochastic gradient descent applied under the quadratic loss, which of course is very convenient for tying together properties of the iterates and gradients. Their main results are essentially composed of two elements: (1) sufficient conditions for the iterates to have a "nice" limiting distribution with tails that are easy to pin down (Thms 2 and 5), and (2) quantitive relationships (via tail index) between different step size strategies, dimensionalities, and batch size settings (all the other Props/Thms besides Thm 1). The key characteristic of this paper is that a rather wide family of step-size strategies are captured (constant, cyclic, uniform, Markov, etc.). The data is assumed to be light-tailed in that all moments are finite, and their strongest results assume Gaussian data.

There are also apparently many additional results in the appendix related to analogous non-asymptotic guarantees, but I have not looked over these, sorry.

**Audience:**

Yes

**Broader Impact Concerns:**

No additional concerns.

**Claims And Evidence:**

Yes

**Requested Changes:**

Please see the comments above; I think there are some valid results here, but the presentation as-is is in my opinion quite sub-par. At the very least, section three needs some serious re-writing to illustrate only the essential differences between current sections 3.1 and 3.2, while eliminating redundancy and unclear, bloated notation.


**Strengths And Weaknesses:**

Overall, my impression is that this paper makes some valid technical contributions related to a machine learning problem of interest to the community, but the clarity and presentation is lacking in several respects. I will try my best to elaborate on the points which are lacking in a constructive but concise fashion below, using some concrete examples from the text.

Taking a macroscopic view of section 3, any reader will notice that sections 3.1 and 3.2 are extremely similar, following a clear template with the only real difference being that the fixed cycle length $m$ in 3.1 is replaced with the stochastic "regeneration times" $r\_{k}$ in 3.2. Quite honestly, this could all be captured in one section, significantly reducing the redundant material, allowing more space for remarks on the technical differences between the two settings (if any). In my opinion, a re-working of this section is critical.

Taking a more microscopic look at the technical material, I find the notation to be extremely bloated with non-essential complexity. Here are a couple obvious examples.

- The superscripts $(\\cdot)^{(m)}$ and $(\\cdot)^{(r)}$ severely reduce readability and increase cognitive load, yet have no explicit impact on the results presented in the main paper. At the very least, I know that the value of $m$ should impact the tail index, but if this impact is only made precise in the appendix, there is absolutely no need to explicitly tack on $m$ to all quantities of interest. Same goes for $r$ (whatever $r$ means; see another comment later on).

- A huge amount of notational complexity is also due to the random batching mechanism $\\Omega\_{k}$. As far as I can tell, this mechanism is essentially meaningless to the analysis, since constant batch $b$ and just one pass is made using iid data. All the notational messiness related to the $\\Omega\_{k}$ guys is equivalent to saying "step sizes are $(\\eta\_{1}/b, \\eta\_{2}/b,\\ldots)$ for some constant $b > 0$". The latter is far simpler, leads to more readable notation, and the link to mini-batching can be easily established with a short remark.

In addition to bloated notation, there is a fair bit of material that is just plain and simply unclear. Here are some examples.

- Thm 1 is riddled with mystery. What is a "non-arithmetic" distribution? What is the difference bounded above by $B^{(m)}$? It looks to be a difference between vectors, but the notation $\\lvert \\cdot \\rvert$ is used; what norm is this? What is $\\lvert M^{(m)}\\rvert$? Determinant? Going beyond these points, is Thm 1 really all that valuable here? I know the authors are trying to constrast the abstract assumptions of Thm 1 with those made in their analysis, but in its present form, I think it does more harm than good.

- There is a bit of a gap in the tail index notions presented in this paper. In section 2 definitions, tail index $\\alpha$ for random vectors is defined using a *constant* $c\_{0}$ free of $u$, but in the main results, the factor depends on $u$ through the mysterious continuous function $e\_{\\alpha}(\\cdot)$.

- In lower bounding the tail index (paragraph before Thm 3), is $h^{(m)}(\cdot)$ monotonic over the positive reals? Or is this why $\\widehat{\\alpha}^{(m)} \\geq 1$ is needed in Thm 3? I think the analysis here could be a lot more transparent. In the present form, readers are left wondering "what's so special about the threshold 1?"

- What is $r$? The authors just say the differences $r\_{k}-r\_{k-1}$ have the same distribution as $r\_{1}$ for all $k$; so is $r$ the common distribution of the differences? If so this is a very poor choice of notation. Furthermore, if $r$ is a random variable (unlike constant $m$ earlier), then one naturally thinks results like Thm 5 should be probabilistic, since for example $e\_{\\alpha^{(r)}}(u)$ would be random if $\\alpha^{(r)}$ is random. I assume this is not the case, with all randomness due to the step sizes being captured in the event $\\{u^{T}x\_{\\infty}^{(r)} > t\\}$, and $r$ is just used here symbolically to differentiate between the setting in 3.1, but this is all quite unclear.

I could go on with similar examples, but I'll get on to one last macroscopic point.

The *title* says "Cyclic and Randomized Stepsizes Invoke Heavier Tails in SGD" and any reader will take this to naturally be a "main finding" of this paper. Here, one naturally wants to ask "heavier than what?". For example, Prop 2 says that cyclic step sizes yield a larger tail index, i.e., *lighter* tails. I assume that in choosing this title, the authors have Prop 1 in mind (by the way, what is $\\rho$ with no superscript?). At this point, I found myself running into the question: "wait, heavier tails are *better*?" There is a now-massive literature on robust stochastic gradient descent, and it is provable that heavy-tailed gradients (typically leading to heavy-tailed iterates) left as-is can never enjoy the strong guarantees that their sub-Gaussian counterparts can (thus justifying clipping and normalization and momentum and such to control the distribution and obtain nice high-probability convergence guarantees). I think a non-trivial fraction of readers of this journal will hear "heavy tails" and assume "worse generalization performance". I know generalization is not discussed directly here (for good reason, I think), but there seems to be a subtext that heavier tails are in general better for learning, which I think is bound to lead to confusion.

Minor points:

- p.1: "instantaneous" loss, not familiar with this term. Intention is "per-instance" sort of notion?
- p.4: "This motivates the study of least square problems ... practical insights to deep learning." I feel like this is probably overstated. While I haven't read the cited papers, if linear least squares is a running assumption, links to most interesting (non-linear) deep learning models vanish at that point. In my opinion, all talk of deep learning should be relegated to the relevant empirical experiments; this paper deals with linear least squares, generalizing recent results on the limiting iterate distribution tail index from constant step sizes to more general step size strategies, nothing more, nothing less.
- p.4: loss function notation is $f\_{i}(x)$ here, doesn't match $f(x,z\_{i})$ given on p.1.

---

### Review · Reviewer_oJUY · 2023-07-15

**Summary Of Contributions:**

This paper considers a general class of Markovian stepseizes in SGD. The authors study the tail-index of the SGD isterates for linear regression and compare the tail-indices among cyclic, Markovian and i.i.d. stepsizes. The authors also compare these stepsizes in both linear regression experiments and deep learning experiments.


**Audience:**

Yes

**Claims And Evidence:**

Yes

**Requested Changes:**

The authors should address all concerns mentioned in the previous part. In addition, the writing can be polished. Some formulas appear more than once in the paper. For example, the updating formula (10) and (16) are the same except the assumption on the H_k and q_k. The authors may reorganize the paper and avoid redundant formulas.

Minor:
What does "MC" in table 1 stand for?

**Strengths And Weaknesses:**

The paper is easy to follow. The studied problem is important and interesting. However, I have following concerns:

1. The theoretical analysis is based on linear regression, which is quite special. Is it possible to generalize the analysis to the general strongly-convex loss functions?

2. Assumption (A3) requires the data comes from zero-mean Gaussian distribution, which in my opinion is a bit strong.

3. SGD usually uses a decreasing stepsize to ensure convergence, e.g., [1,2]. However, this paper does not compare the Markovian stepsizes with the commonly used decreasing stepsizes.

4. In the experimental part, I think the authors should perform experiments on convolutional neural networks such as the VGG networks rather than the simple 3-layer fully connect networks.

[1] Sebbouh, Othmane, Robert M. Gower, and Aaron Defazio. "Almost sure convergence rates for stochastic gradient descent and stochastic heavy ball." Conference on Learning Theory. PMLR, 2021.
[2] Gower, Robert Mansel, et al. "SGD: General analysis and improved rates." International conference on machine learning. PMLR, 2019.

---

### Decision · Action_Editors · 2023-08-18

**Recommendation:** Accept with minor revision

**Comment:**

The reviewers' opinion on this paper was split, with a majority voting for accept.

A main concern of the reviewers was that the setting (linear regression) and some assumptions (Gaussian data) may be too restrictive to draw conclusions for Deep Learning practice from the results of this paper. In the rebuttal phase, the authors added experimental results to shed more light on this gap, and presented slightly generalized theorem statements. Overall, this paper makes some progress in deepening our understanding of (non-constant) Markovian step sizes in SGD and these results may be of interest to the TMLR community.

Minor revision: please upload a CR version without color annotations.

**Audience:**

Yes, this topic is of interest to parts of the TMLR audience.

**Claims And Evidence:**

The mathematical claims made in this paper are derived under a set of assumptions that are clearly stated.

After some clarification by the authors in the discussion period, the reviewers were satisfied with the accuracy of the claims.